

# Triple negative breast cancer cells acquire lymphocyte proteins and genomic DNA during trogocytosis with T cells

Anutr Sivakoses[1,2], Haley Q. Marcarian[1,2], Anika M. Arias[1],
Alice R. Lam[3,4], Olivia C. Ihedioha[2], Juan A. Santamaria-Barria[5],
Geoffrey C. Gurtner[6] and Alfred L. M. Bothwell[2,7]

[1] Arizona Cancer Center, University of Arizona, Tucson, Arizona, United States
[2] Department of Pathology, Microbiology, and Immunology, University of Nebraska Medical Center, Omaha, Nebraska, United States
[3] Department of Biophysics, Stanford University School of Medicine, Palo Alto, California, United States
[4] Department of Biology, Stanford University, Palo Alto, California, United States
[5] Department of Surgery, University of Nebraska Medical Center, Omaha, Nebraska, United States
[6] Department of Surgery, University of Arizona, Tucson, Arizona, United States
[7] Department of Immunobiology, Yale University, New Haven, Connecticut, United States

Corresponding author
Alfred L. M. Bothwell,
albothwell@unmc.edu

## ABSTRACT

Trogocytosis is the process by which a recipient cell siphons small membrane fragments and proteins from a donor cell and can be utilized by cancer cells to avoid immune detection. We observed lymphocyte specific protein expressed by triple negative breast cancer (TNBC) cells *via* immunofluorescence imaging of patient samples. Image analysis of Cluster of Differentiation 45RA (CD45RA) expression, a naïve T cell specific protein, revealed that all stages of TNBCs express CD45RA. Flow cytometry revealed TNBC cells trogocytose CD45 protein from T cells. We also showed that the acquisition of these lymphoid markers is contact dependent. Confocal and super-resolution imaging further revealed CD45+ spherical structures containing T cell genomic DNA inside TNBC cells after co-culture. Trogocytosis between T cells and TNBC cells altered tumor cell expression of *PTPRC*, the gene that encodes for CD45. Our results revealed that CD45 is obtained by TNBC cells from T cells *via* trogocytosis and that TNBC cells express CD45 intracellularly and on the membrane.

## INTRODUCTION

Immune cells are derived from hematopoietic stem cells (HSCs) and express a specific subset of proteins such as CD45, CD56, CD14, and CD16 not found in epithelial and mucosal cells (*Marcarian et al., 2024*). However, some tumor cells acquire immune cell proteins *via* trogocytosis (*Hasim et al., 2022*; *Shin et al., 2021*). Trogocytosis is the process by which an acceptor cell "nibbles on" and obtains small fragments of the lipid membrane and membrane protein from a donor cell (*Joly & Hudrisier, 2003*). Transfer of protein due

to trogocytosis has been characterized in mice as a mechanism for antigen presentation between conventional dendritic cells and T cells (*Shin et al., 2021*).

We have previously shown that colon cancers acquire immunoregulatory proteins from infiltrating lymphocytes through trogocytosis (*Shin et al., 2021*). *In vitro* and *in vivo* colorectal cancer organoid models demonstrated trogocytic interactions between tumor and immune cells, resulting in the acquisition of immunoregulatory proteins CTLA4, VISTA, TIM3, CD38, and CD80 (*Shin et al., 2021*). Further studies concluded that natural killer (NK cells) may also obtain immunoregulatory protein PD-1 from double-positive PD-1/PD-L1 leukemia cells leading to a dampened NK cell response (*Hasim et al., 2022*; *Shin et al., 2021*). Leukemia cells also transfer immunoregulatory proteins to CAR-T cells, which reduced immunotherapy efficacy (*Gurung et al., 2021*). Taken together, these findings demonstrated that trogocytosis of immune cell proteins by tumor cells can attenuate the immune response to a tumor.

Luminal B (LumB) breast cancers have previously been shown to also undergo trogocytosis with NK cells, resulting in the transfer of Human Epidermal Growth Factor Receptor 2 (HER2) from breast cancer cells to Natural Killer (NK) cells (*Suzuki et al., 2015*). Although trogocytosis has been observed in HER2+ breast cancers, the transfer of immune cell proteins to breast cancer cells has not yet been characterized.

For this study, we focused on triple negative breast cancer (TNBC) as this subtype contains the highest density of tumor infiltrating lymphocytes (TILs), including macrophages and T cells, and may display the most trogocytosis between immune and cancer cells (*Harbeck & Gnant, 2017*). TNBCs do not express any of the three major hormone receptors (ER, PR, and HER2) found in other subtypes of breast cancer and are considered the most aggressive subtype of breast tumors due to the lack of therapeutic targets (*Harbeck et al., 2019*). We first investigated the expression of CD45 on breast cancer cells by quantifying the frequency of their expression *via* immunofluorescence staining of formalin-fixed paraffin-embedded (FFPE) slides. Acquisition of CD45 by tumor cells from T cells *in vitro* was also assessed by flow cytometry, confocal imaging, and structured illumination microscopy (SIM) analysis was also investigated. Tumor cell acquisition of genomic DNA (gDNA) originating from primary T cells was also visualized. The overall aim of this study was to quantify and visualize the transfer of immune cell proteins and gDNA to TNBC cells.

## MATERIALS AND METHODS

### Cell culture

Immortalized cell lines of triple negative breast cancers HCC1937, MDAMB231, and MDAMB436 were provided to our laboratory courtesy of Dr. Joann Sweasy at the University of Nebraska Medical Center and Dr. Karen Anderson at Arizona State University. Portions of this text were previously published as part of a preprint https://doi.org/10.1101/2024.08.09.607029 (*Sivakoses et al., 2024*). TNBC cell lines were cultured according to ATCC recommended guidelines. Human Primary T cells were obtained from the Elutriation Core Facility at the University of Nebraska Medical Center (UNMC) and were cultured in X-VIVO 15 Serum-Free Hematopoietic Cell medium (02-053Q; Lonza,

Basel, Switzerland) supplemented with 20 ng/mL recombinant human IL-2 and 7 μL/mL of ImmunoCult™ Human CD3/CD28/CD2 T Cell Activator (10970; Stem Cell Technologies, Vancouver, Canada) every week.

## Immunostaining of FFPE tissues

Paraffin-embedded tissue slides of malignant stage I-IV TNBC tissue were obtained from the Tissue Acquisition and Cellular/Molecular Analysis (TACMASR) core facility at The University of Arizona and BioMax (BioMax BR1202a). The University of Arizona eIRB committee reviewed and determined that our studies was assigned the determination of "Not Human Research" (STUDY00000912). Antibodies against CD45RA, CD14, CD56, and Pan-Cytokeratin I/II were used in this portion of the study. Slides were treated with xylene for deparaffinization and washed in decreasing ethanol concentrations. Antigen retrieval was conducted by treatment with a sodium-citrate based Antigen Unmasking Solution (H-3300; Vector Laboratories, Newark, CA, USA) at boiling temperatures for 10 min and left to cool for an additional 20 min. Slides were then blocked with a 20% donkey serum reconstituted in 0.1% PBS-Tween for 30 min and then incubated with primary antibody overnight at 4 °C. Cells were then washed with 0.1% PBS-Tween 3 times at 5 min each before labeling with secondary antibodies for 1 h at room temperature. Samples were quenched using an autofluorescence quenching kit for 3–5 min. Coverslips were mounted on specimen slides using Fluoromount-G (00-4958-02; Invitrogen, Waltham, MA, USA) and left to dry for 24 h prior to imaging on an ECHO Revolution WF fluorescence microscope.

## Image analysis of CD45RA

Our lab has developed TrogoTracker, an ImageJ-based (FIJI) macro. Images of FFPE samples were captured using the 60X objective on the ECHO Revolution microscope (*Marcarian et al., 2024*). Acquired images were imported into FIJI and single channel images were generated on separate windows. Nuclei were located using the StarDist2D FIJI package to generate a region of interest (ROI) with individual nuclei. A Voronoi thresholding algorithm was then used to identify tumor cell borders indicated by PanCk I/II staining and then overlaid onto the ROI determined by the nuclei channel to generate a new ROI that only accounts for nuclei that are encapsulated by the tumor cell border, thereby isolating only tumor cells as the only type of cells in the image to be quantified. The channel containing CD45RA is then processed by a background subtraction algorithm and normalization *via* a CLAHE histogram prior to being converted into a binary image. The surface area of each tumor cell is then measured and only tumor cells containing trogocytic markers that constitute between greater than 55% and less than 90% of their total surface area were flagged as a trogocytic cell. These cutoffs were considered conservative as it discards cells that may have autofluorescence and very bright fluorescent anomalies such as paraffin debris. Automation of TrogoTracker can be conducted using a MatLab/Python wrapper to automate image analysis. The original version of TrogoTracker (bit.ly/TrogoTracker) and a newer, more automated revision to TrogoTracker (bit.ly/TrogoTrackerRev1) can be found on our lab GitHub repository (github.com/bothwelllab).

## Flow cytometry of *in vitro* co-cultures

Primary T cells were treated with EdU (C10337; Invitrogen, Waltham, MA, USA) or nothing for 16 h prior to co-culture. Tumor cells were seeded into a six-well plate 1–2 h prior to addition of T cells. Primary T cells were then added at a 5:1 ratio compared to cancer cells and allowed to either co-culture freely or while separated by a 0.3 µm transwell for 16 h. T cells were aspirated from co-cultures prior to trypsinization of tumor cells. Tumor cells were then trypsinized, pelleted, and washed with PBS. Trypsinization was stopped using DMEM containing 10% FBS. In EdU co-cultures, fluorescent labeling of tumor cells was prepared according to the kit directions (C10337; Invitrogen, Waltham, MA, USA). Cells were then resuspended in a 100 µL solution of PBS containing an antibody targeting human CD45 conjugated to PE-Cy7 (25-9459-42; Invitrogen, Waltham, MA, USA) and incubated for 30 min in the dark at room temperature. Samples were then washed by centrifugation and with PBS 2 times and then labeled with 7AAD viability dye (S10349; Invitrogen, Waltham, MA, USA) on ice 30 min prior to running samples on an BDFACS LSR II flow cytometer. Since the TNBC cells were much larger than the primary T cells, we initially drew our gates based on size using FSC-A/SSC-A discrimination and then further gated on only BFP-expressing tumor cell singlets to exclude any T cells that were not washed out.

## Real-time, quantitative PCR analysis of *in vitro* co-cultures

Co-cultures were set up according to the methods sections outlined above and were sorted on the BDFACS AriaII fluorescent cell sorter on ice. Cells within the CD45$^{bright}$ tumor cell gate were retrieved and RNA was extracted using the Qiagen RNAeasy Plus Micro kit (74004; Qiagen, Hilden, Germany) using the provided kit instructions as the experimental group. RNA was quantified on a Nanodrop to validate concentration and purity. cDNA was prepared using Applied Biosystems High-Capacity cDNA Reverse Transcriptase kit containing reverse transcriptase and randomized primers (4368814; AB, Waltham, MA, USA). cDNA was stored at −20 °C prior to quantitative PCR analysis. Primer sequences were obtained previously published results with the following sequences: CD45-F 5′-CTTCAGTGGTCCCATTGTGGTG-3′, CD45-R 5′-CCACTTTGTTCTCGGCTTCCAG-3′ (NCBI Gene ID 5788; Product length 63 bp). Primer specificity was verified using NCBI BLAST and SnapGene v7.0. For quantitative analysis, template cDNA and 10 mM primers were intercalated with fluorescent BioRad iTaq SYBR Green containing Mg$^{2+}$ and dNTPs (1725120; BioRad, Hercules, CA, USA). Samples were prepped to a final of 20 uL per well in technical triplicates and RT-qPCR analysis was run on an Applied Biosystems QuantStudio3 using the suggested parameters specific to the iTaq SYBR provided by BioRad. Melt curve analysis was validated and conducted using QuantStudio Software. Non-template controls were also included and had no detectable Cq value. No Cq values above 35 were considered for analysis. Fold change was calculated by comparing the Cq of *PTPRC* to housekeeping gene *GAPDH* in CD45$^{bright}$ tumor cells to monocultured tumor cells as a control. *GAPDH* was selected as a housekeeping control due to its high expression in all cell lines used in RT-qPCR analysis, making it an appropriate comparison marker across all of our experimental and control groups. Data was exported from the

ThermoFisher Connect Platform, analyzed, and prepared for final publication on Microsoft Office 365 Excel and Graphpad Prism 10.

## Immunofluorescence imaging of *in vitro* co-cultures

For visualization of T cell membrane fragments, primary T cells were treated with DiD dye (V22887; Invitrogen, Waltham, MA, USA) for 25 min at 37 °C. Cancer cells were plated and allowed to attach overnight on glass-bottomed MatTek dishes (P35G-1.5-14-C; MatTek, Ashland, MA, USA) prior to addition of T cells. T cells were added at a 10:1 ratio compared to cancer cells and allowed to co-culture with cancer cells for 4, 8, or 16 h. Cells were then fixed with 4% paraformaldehyde for 15 min at room temperature and quenched with a 50 mM ammonium chloride ($NH_4Cl$) solution for 5 min at room temperature. For EdU co-cultures, cells were labeled with an Alexa Fluor 647-conjugated fluorescent azide that binds to EdU (C10337; Invitrogen, Waltham, MA, USA). To preserve membrane integrity, blocking was conducted using a 20% donkey serum solution reconstituted in 0.1% PBS-Saponin for 30 min at room temperature. Cells were then incubated with primary antibody overnight at 4 °C. Secondary antibodies, Hoechst (DAPI), and a phalloidin antibody conjugated to Alexa Fluor 568 were added after primary antibody labeling for 1 h at room temperature. Images were captured on the ECHO Revolution, Zeiss LSM710/800 Confocal, and Zeiss Elyra PS.1 Super Resolution microscope. The diameters of trogosomes were quantified using ImageJ.

## Lentiviral transfection and infection of mammalian cell lines

HEK293T cells were obtained from the American Type Culture Collection (ATCC) and cultured in DMEM with 10% FBS. The pCI GFP-H2B vector was a generous gift from Dr. Ghassan Mouneimne at the University of Arizona. For transfections, HEK293T cells were plated at 70% confluency. 30 μg of polyethylenimine (PEI) was then combined with 250 μL of OptiMEM (31985062; Gibco, Waltham, MA, USA) and incubated for 5 min at room temperature in a 1.5 mL Eppendorf tube. In a separate 1.5 mL tube, 5 μg of pCI H2B-GFP, 3.75 μg of psPAX2 packaging plasmid, and 1.25 μg of pMD2.G expression vector was combined with an additional 250 μL of OptiMEM. Both tubes consisting of the PEI/OptiMEM and plasmid DNA/OptiMEM were then combined, vortexed, and allowed to incubate for 20 min at room temperature. The transfection solution was then pipetted onto the HEK293T cells and incubated in a tissue culture incubator (37 °C, 5% $CO_2$) overnight. Media was changed 16 h after initial transfection and then collected at 48- and 72-h post-transfection. Collected media was then centrifuged to remove residual cell debris and passed through a 0.4 μM syringe filter. Media containing the lentivirus was then used to immediately infect T cells or frozen at −80 °C for long-term storage.

# RESULTS

## Triple negative breast cancer cells express CD45RA

Although trogocytosis has only been observed in Luminal B breast cancer to date, transfer of immune cell proteins to cancer cells have been established in other cancers, including colorectal cancer and leukemia (*Hasim et al., 2022*; *Shin et al., 2021*). We hypothesized that

immune cell proteins are acquired and expressed by breast cancer cells through trogocytosis. We first visualized the expression of immune-cell specific markers on FFPE TNBC tissue. Pan-cytokeratin I/II (pan-Ck I/II) was used to define breast tumor cell borders. FFPE slides were immunolabeled with antibodies against immune cell markers CD45RA, CD14, CD56, and CD16. Upon visualization using a fluorescent microscope, we discovered all four immune cell markers expressed on TNBC cells (Fig. 1A; Fig. S1).

We then focused specifically on CD45RA due to the high expression seen in our initial imaging and immunostained a tumor microarray (Tumor Microarray and FFPE slides of primary tumors; $n$ = 52). Utilizing TrogoTracker v1.1, a FIJI-based macro developed by our lab, we recorded the fluorescence intensity of the CD45RA channel expressed by the tumor cells. The CD45RA expression of our tumor samples was then compared against a secondary antibody-only control to determine the percentage of tumor cells that were expressing CD45RA. Although there were no discernable differences with regards to CD45RA expression across tumor stages, 30–60% of individual tumor cells in each sample expressed CD45RA (Fig. 1B). To further understand whether expression of CD45RA was localized to small compartments of tumor tissue or whether expression could be seen throughout the whole sample, we acquired large image stitches of tumor samples and discovered that expression of CD45RA was consistent in all sites containing tumor tissue (Fig. 1C). We then imaged these TNBC samples on a confocal microscope to validate and exclude any false-positive CD45RA labeling caused by an immune cell being directly above or below a tumor cell (Fig. 1D). Taken altogether, we demonstrated that expression of CD45RA is common across all stages of TNBCs.

## Trogocytosed CD45 protein by triple negative cancer cells alters *PTPRC* expression

Due to the frequency of CD45RA expression in our FFPE TNBC samples, we then elucidated the origin of CD45 protein expression *in vitro* and co-cultured TNBC cell lines (MDAMB231, MDAMB436, and HCC1937) with CD3$^+$ T cells derived from healthy donor PBMCs. Immortalized TNBC cell lines were transduced with lentiviral particles expressing Blue Fluorescent Protein LifeAct (BFP-LifeAct), a small 17 amino acid peptide that binds filamentous actin. BFP-LifeAct was used as an internal marker to identify our tumor cells as T cells do not acquire BFP-LifeAct after co-coculture with TNBC cells (Fig. S2A).

TNBC cells were co-cultured with CD3$^+$ T cells at a 1:5 ratio for 16 h and expression of CD45 was assessed by flow cytometry. Between 15% and 30% of TNBC cells expressed CD45 after co-culture, compared to 0% expression when co-cultured tumor and immune cells were separated by a transwell insert (Fig. 2A; Fig. S2B). Interestingly, while we found two subpopulations of CD45$^{negative}$ and CD45$^{bright}$ tumor cells, we also found an intermediate population of CD45$^{dim}$ tumor cells (Fig. 2B). We then quantified the average mean fluorescent intensity (MFI) of each population and discovered that co-cultured tumor cells were 5 to 17-fold higher fluorescence intensity compared to co-cultures separated by transwells (Fig. 2B). Interestingly, HCC1937 TNBC cells did not trogocytose as much as MDAMB231 and 436 (Fig. S2C).

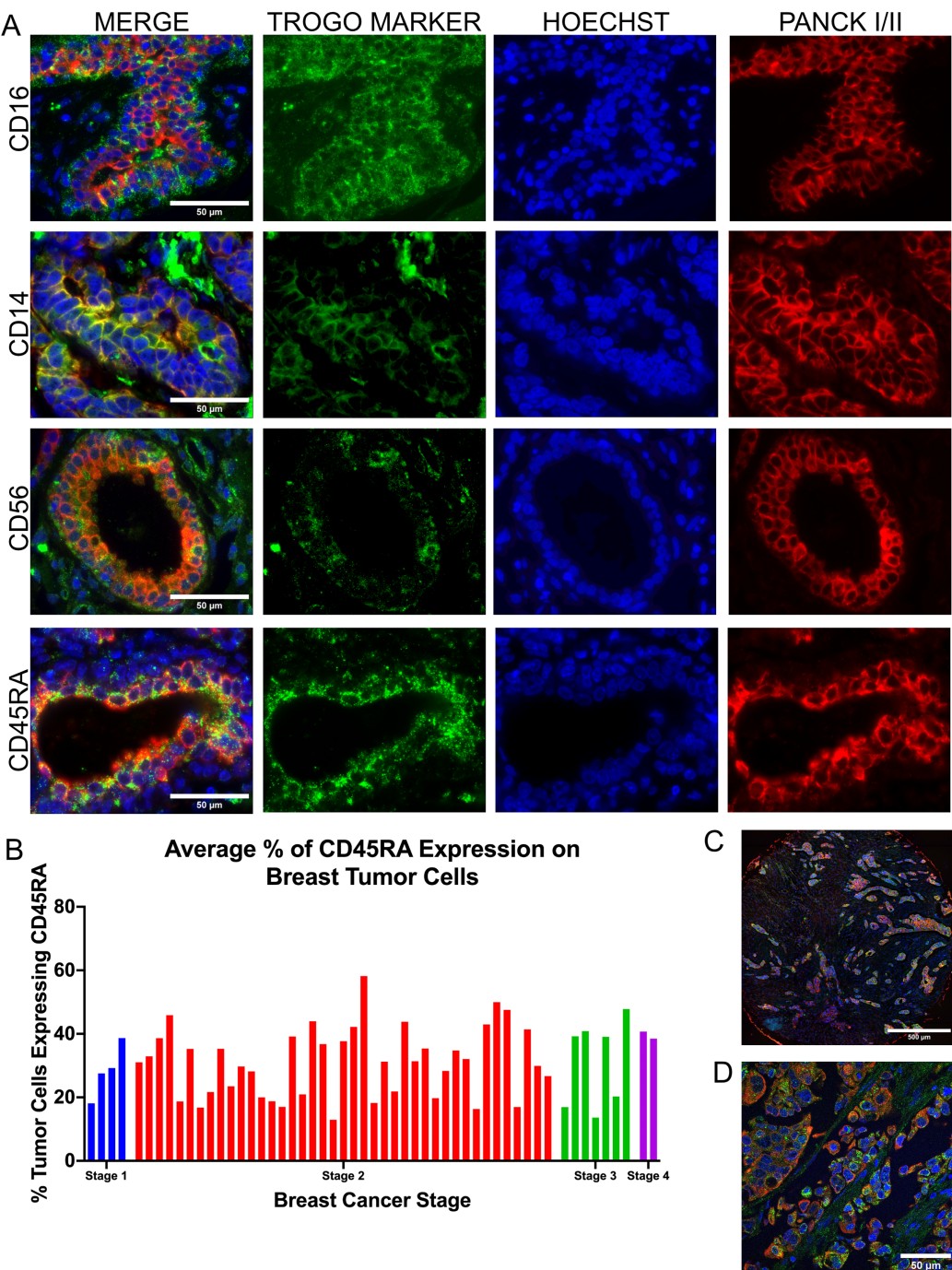

**Figure 1 Triple negative breast cancer cells express lymphocyte proteins.** (A) Widefield fluorescence imaging (Olympus Plan X Apo Oil Phase NA 1.42 WD 0.15 mm) of immune cell proteins CD14, CD16, CD56, and CD45RA (green) expressed on tumor cells labeled by pan-cytokeratin I/II (Red). Hoechst labeling nuclei is displayed in blue. (B) Quantification of the percentage of tumor cells expressing CD45RA across 54 patient samples and organized by tumor stage. (C) Large image stitch of tumor cells expressing CD45RA (green) in different sites. (D) Confocal imaging of an intermediate Z-layer of CD45RA-expressing tumor cells (green).

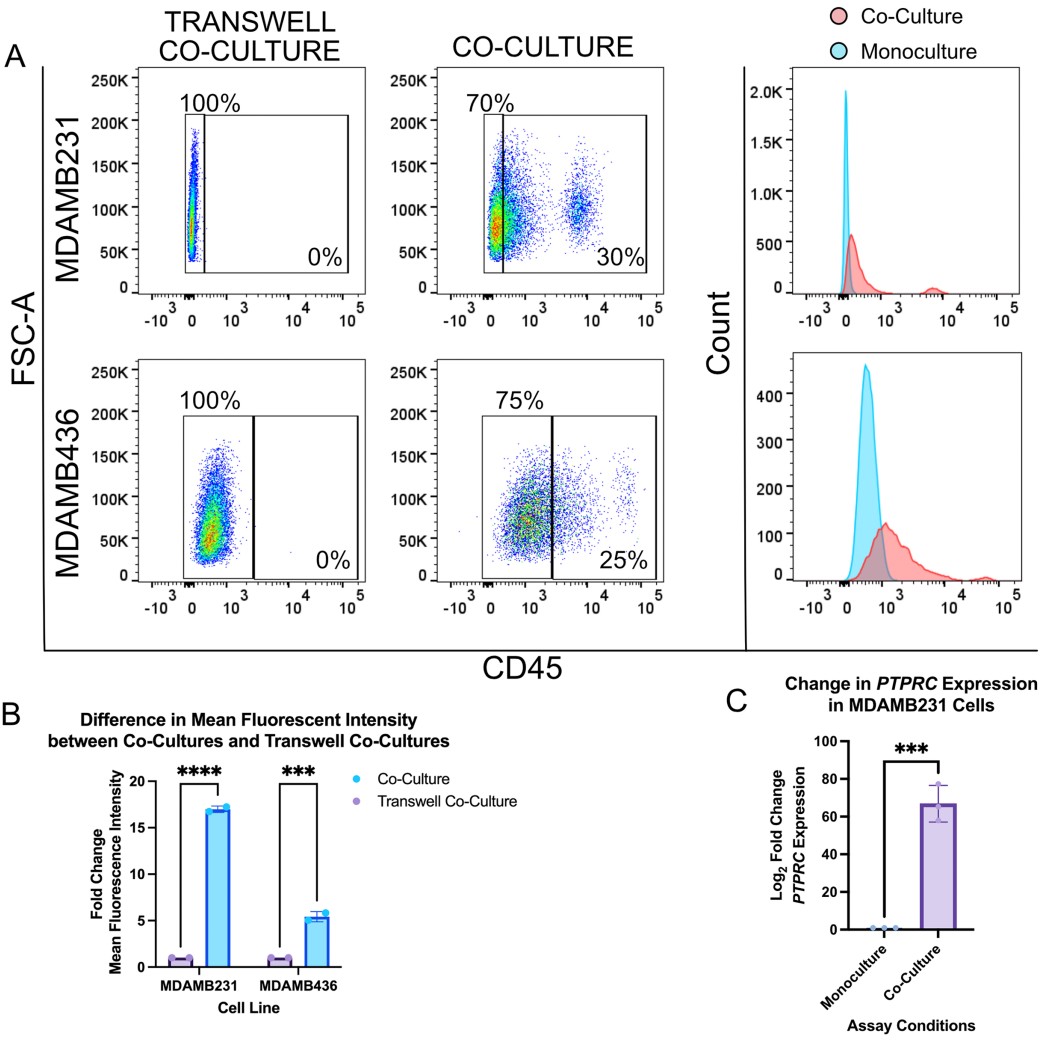

**Figure 2** **CD45 is transferred from CD3+ T cells to TNBC cells.** (A) Flow cytometry analysis of CD45 expression on tumor cells. Tumor cells were gated based on size, singlets, viability, and BFP-LA expression. Histograms show the shift in fluorescence between co-cultures (red) and transwell co-cultures (blue). (B) Fold change in the mean fluorescence intensity of co-culture tumor cells compared against transwell co-cultures. A two-way ANOVA statistical analysis and a Bonferroni *post hoc* test was conducted to analyze the data and determine statistical significance ***$p = 0.0002$, ****$p < 0.0001$. (C) Fold change of differentially expressed CD45 in co-cultured and transwell co-cultured tumor cells. An unpaired t test was performed to analyze the data ***$p = 0.0003$.

We then investigated whether the expression of the CD45 gene, *PTPRC*, was altered. We set up co-cultured BFP-LifeAct[+] MDAMB231 cells and CD3[+] primary T cells and isolated double-positive (BFP-LifeAct[+]CD45[+]) MDAMB231 cells *via* fluorescence-activated cell sorting (FACS). RNA was then extracted from sorted cells and the difference in expression of *PTPRC* by monocultured and trogocytic MDAMB231 cells was assessed by a real-time quantitative PCR expression. We then compared to the expression of *PTPRC* in CD3[+] T cells to CD45[+] MDAMB231 cells and found that expression was on average two-fold higher in T cells compared to CD45[+] MDAMB231 cells (Fig. S2D). In contrast, CD45[+]

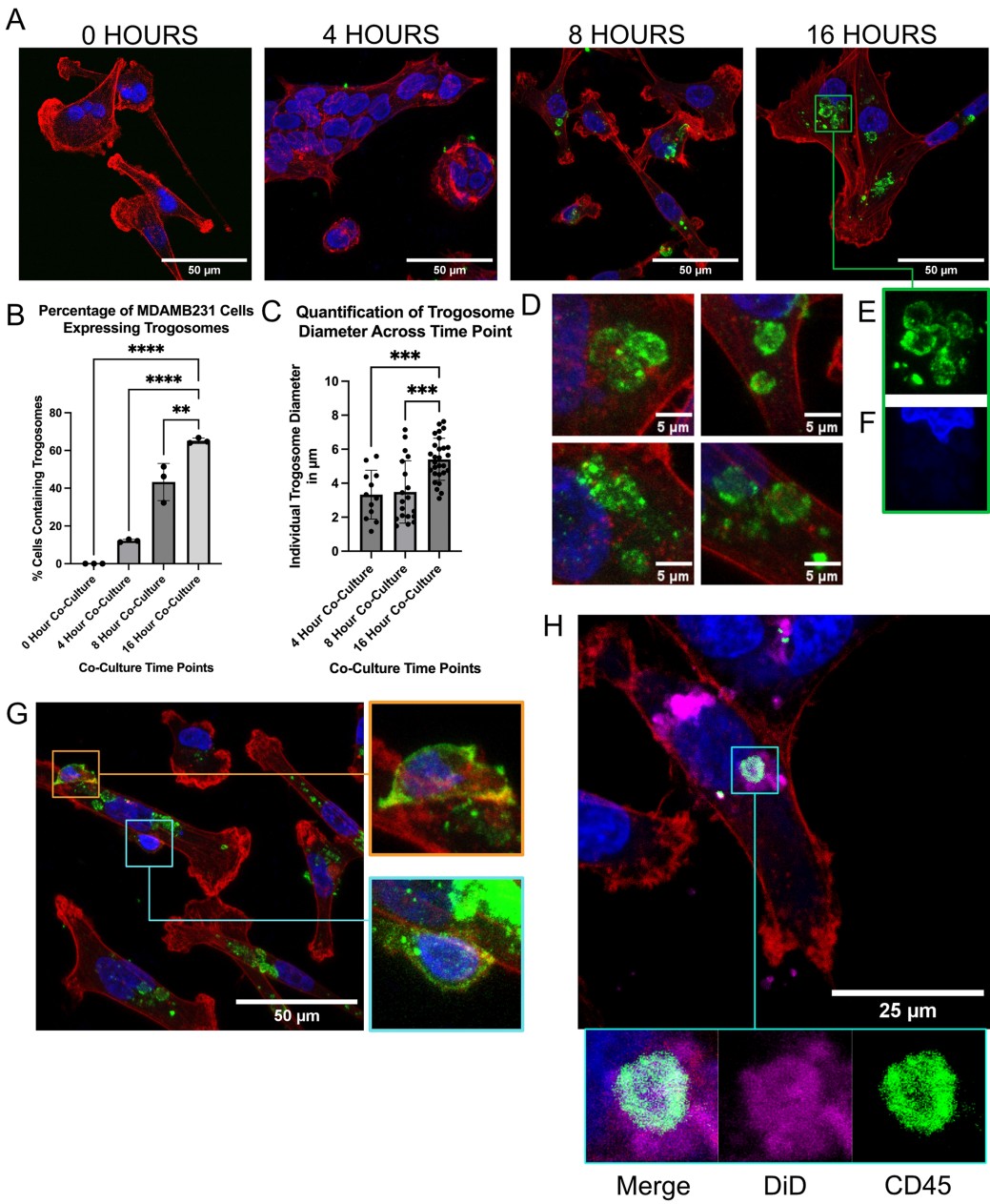

**Figure 3 TNBC cell lines acquire trogosomes from CD3+ primary T cells.** (A) T cells and TNBC cells were cocultured and fixed and immunolabeled for CD45 (green), phalloidin (red), and nuclei (blue) at different timepoints post co-culture. (B) Percentage of cells containing trogosomes larger than 5 m, per timepoint (C and D) Quantification of average trogosome size. An ordinary one-way ANOVA and a Bonferroni *post hoc* statistical test was performed to analyze the data **$p$ = 0.0030, ***$p$ = 0.0003, ****$p$ < 0.0001. (E) Closeup of hollow trogosomes and altered nuclei morphology. An ordinary one-way ANOVA and Bonferroni *post hoc* statistical test was performed to analyze the data ***$p$ = 0.0003, ****$p$ < 0.0001. (G) Visualization of CD45 hi T cell (orange box) and a T cell with depleted CD45 (teal box; brightness increased for visualization). (H) Visualization of DiD membrane dye (magenta), Phalloidin (red), CD45 (green), and Nuclei (blue).

MDAMB231 cells had an average of a 60-fold change increase over CD45⁻ MDAMB231 cells (Fig. 2C). Taken together, we demonstrated that trogocytosis may involve the transfer of nucleic acids such as RNA.

## TNBC cells acquire intracellular CD45 trogosomes from T cells

Next, we investigated the phenotype of CD45 expression within our TNBC cell lines. We performed co-cultures for 4, 8, and 16-h prior to confocal imaging. Interestingly, we discovered the presence of multiple three-dimensional spheroids inside tumor cells that we refer to as trogosomes in our co-cultures (Fig. 3A; Figs. S3A–S3C). Trogosomes were not present in the monocultured cells (Fig. 3A). When the number of cells containing trogosomes was quantified and compared to the number of cells containing no trogosomes, we discovered that longer co-culture durations led to increased trogosome acquisition by the tumor cells, with a 16-h co-culture resulting in over 60% of TNBC cells acquiring CD45 (Fig. 3B; $n = 50$ cells per replicate, per time point).

At 4-h post co-culture, MDAMB231 cells began to acquire trogosomes that were smaller in size compared to 8-h co-cultures (Figs. 3A–3C). Further investigation of our 4 and 8-h co-cultures revealed fewer trogosomes larger than 5 μm in diameter relative to the 16-h co-cultures (Fig. 3C). Beginning at the 8-h co-culture, we identified the formation of large trogosomes that measured 5 μm in diameter (Fig. 3D). Trogosomes were also much larger than the conventional sizes of exosomes or lysosomes, which on average range between 0.1 and 1.2 μm (*de Araujo et al., 2020*; *Gurung et al., 2021*). The majority of trogosomes were hollow on the inside (Figs. 3D and 3E) but still altered the shape of the cancer cell nuclei (Fig. 3F), demonstrating that they are situated within the tumor cell rather than above or below. We also identified two T cells that were adhered to a tumor cell (Fig. 3G). Interestingly, one T cell expressed more CD45 (Fig 3G; orange box) than the other T cell (Fig. 3G; teal box), suggesting that CD45 expressed by a T cell can be depleted by trogocytosis. Due to their spherical structure, we then hypothesized that these trogosomes were coated by membrane lipids originating from T cells. We repeated the 16-h co-culture and labeled primary T cells with DiD dye, a lipophilic carbocyanine dye, prior to co-culture and visualized T cell membrane fragments that entered the tumor cell. We found that these DiD dye bubbles colocalized with only some of the CD45 trogosomes (Fig. 3H).

As we previously observed that these trogosomes altered the shape of the tumor cell nucleus, we investigated any changes in subcellular structures surrounding the trogosome by utilizing the super-resolution microscopy technique structured illumination microscopy (SIM). We utilized SIM to surpass diffraction limits of conventional imaging and further understand any interactions between the trogosome and the subcellular structures of the tumor cell. We found that there was a hollow cavity in the actin surrounding our trogosomes in TNBC cells (Fig. 4A; Fig. S4A). To validate that this was not a T cell within the tumor cell, we generated a three-dimensional projection using a Z-series capture and visualized the cell from a side perspective (Fig. 4B). We then observed that the trogosome is surrounded by actin filaments but contained minimal actin inside the trogosome (Fig. 4C). Interestingly, we found Hoechst, a DNA marker dye, within this

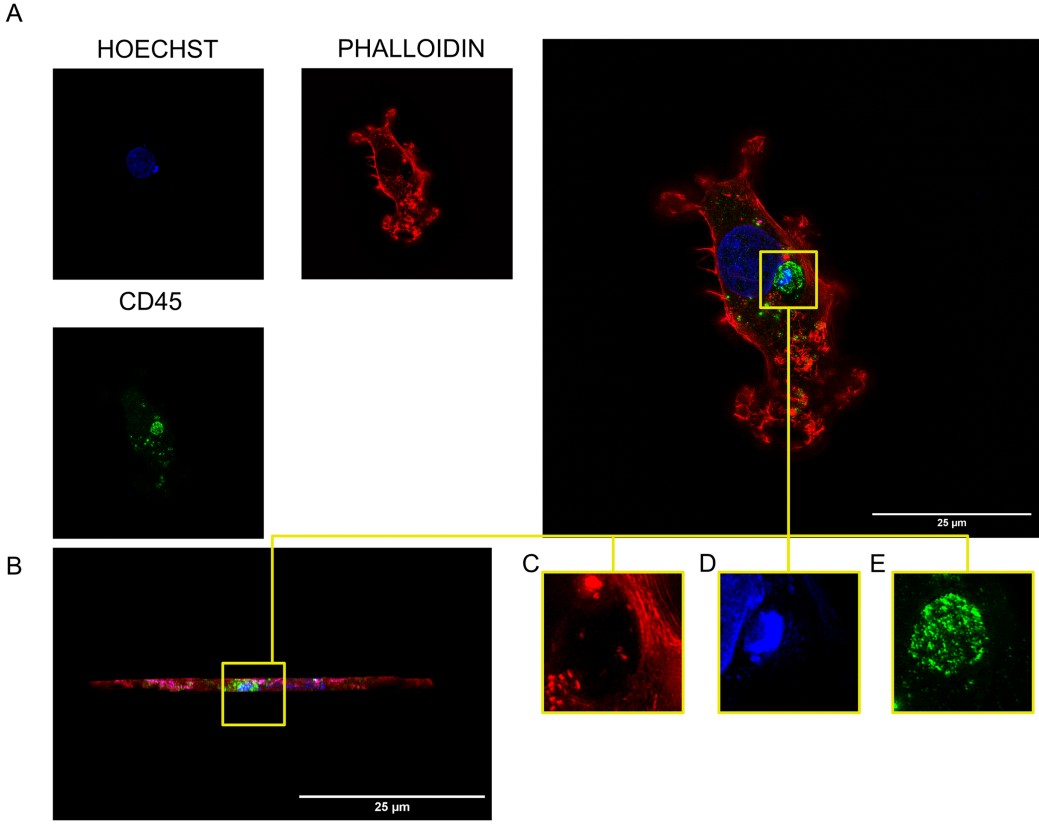

**Figure 4** **Visualization of DNA within a trogosome.** (A) Visualization of MDAMB231 cell containing a trogosome labeled for nuclei (blue), CD45 (green), and Phalloidin (red) using Structured Illumination Microscopy. (B) Side profile of MDAMB231 cell containing trogosome. (C) Enhanced view of phalloidin labeling. (D) Enhanced view of nuclei labeling inside the trogosome. (E) Enhanced view of phalloidin encapsulating the trogosome.

trogosome (Fig. 4D). We demonstrated that the DNA is encapsulated in the trogosome (Fig. 4E).

## TNBC cells acquire gDNA from T cells

Due to the observation of Hoechst inside the trogosome, we then investigated whether TNBC cells obtained gDNA from T cells during trogocytosis. Using the pre-established co-culture models for flow cytometry and imaging, we labeled primary T cells with 5-Ethynyl-2′-deoxyuridine (EdU), a thymidine analog used as a marker for *de novo* DNA synthesis. T cells were co-cultured with MDMAB231 cells for 16-h and then labeled with a fluorescent azide to bind to EdU. Our results show that between 7% to 10% of our co-cultured TNBC cells were labeled positively with EdU that were not present in our monocultured and transwell co-culture controls (Fig. 5A).

We then infected CD3+ cells with lentivirus containing GFP-tagged Histone-H2B (GFP-H2B). After co-culture with immune cells, 3% of MDAMB231 cells expressed GFP-H2B protein compared to transwell co-cultures (Fig. 5B), indicating that protein transfer due to trogocytosis is not exclusive to membrane proteins and can include nuclear proteins. Interestingly, we identified a small subset of CD45⁻GFP-H2B⁺ trogosomes that

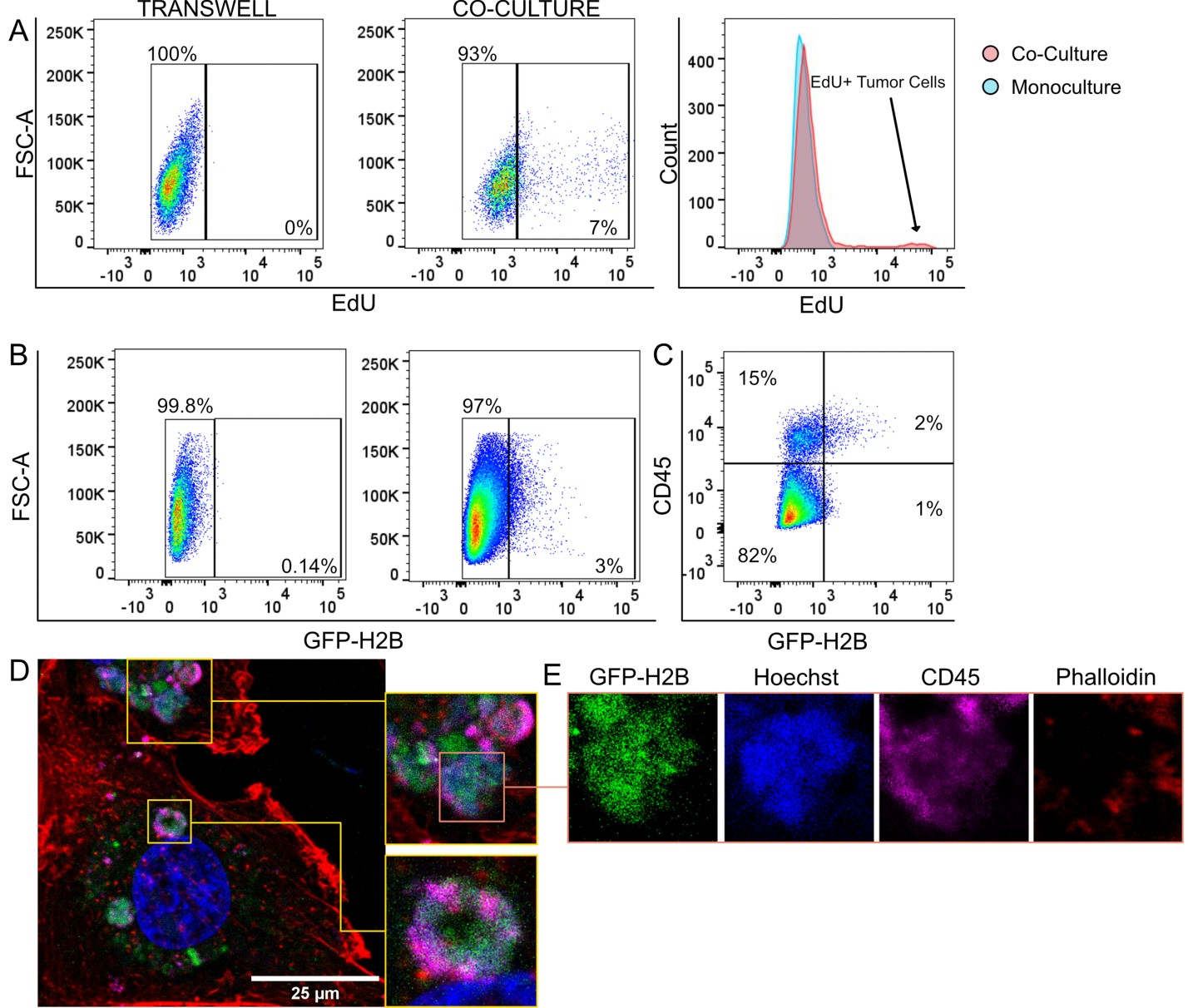

**Figure 5 Tumor cells acquire gDNA from T cells.** (A) Flow cytometry assay of MDAMB231 cells co-cultured with EdU-labeled CD3+ T cells. (B) Flow cytometry analysis of MDAMB231 cells co-cultured with H2B-GFP CD3+ T cells. (C) Flow cytometry plot showing four populations of CD45-H2B-GFP- (lower left gate), CD45+ H2B-GFP- (upper left gate), CD45+ H2B-GFP+ (upper right gate), CD45- GFPH2B+ (lower right gate) MDAMB231 cells. (D) Visualization of trogosomes within a MDAMB231 cell containing CD45 (magenta), Hoechst dye (blue), and GFP-H2B (Green). (E) Single channel images of single trogosome containing GFP-H2B, Hoechst dye, and CD45.

were not coated CD45 (Fig. 5C). Co-cultures between GFP-H2B T cells and MDAMB231 cells were then repeated and imaged with a confocal microscope (Fig. 5D). We discovered the presence of CD45$^+$GFP-H2B$^+$ and CD45$^-$GFP-H2B$^+$ trogosomes, suggesting that the H2B-GFP may be trafficked in with other membrane fragments independent of CD45 transfer (Fig. 5D; Fig. S4B). Additionally, we also visualized H2B-GFP co-localized with small Hoechst micronuclei, further suggesting gDNA transfer (Figs. 5D and 5E; orange

box). We observed the transfer of both gDNA and proteins from T cells to tumor cells, which may lead to a further altered transcriptome through increasing the tumor mutational burden.

## DISCUSSION

We first utilized TrogoTracker v1.1, an ImageJ/FIJI-based macro developed by our lab to quantify CD45RA expression in FFPE histology slides and revealed that every tumor sample we analyzed expressed CD45RA in a cohort of 54 TNBC tumors. CD45RA expression was also consistently expressed in every tumor stage, although there was no significant difference in the expression between each stage. Our data suggests that trogocytic interactions between tumor and immune cells occur in early tumor stages and sustained lymphocyte protein expression may be due to subsequent trogocytic interactions or the trogocytosed lymphocyte DNA being transcribed.

*In vitro* co-cultures between TNBC cell lines and $CD3^+$ primary T cells corroborated that CD45 was transferred from T cells to tumor cells upon contact. Although this study only evaluated transfer of CD45 between T cells and TNBC cells, our FFPE data suggest that TNBC cells may also acquire CD45 from macrophages, NK cells, and other types of immune cells as evidenced by expression of immune cell markers CD14, CD56, and CD68 on tumor cells. Co-cultures also revealed that trogosomes composed of membrane fragments and CD45 protein were acquired by tumor cells after co-culture with T cells. The trogosomes found within TNBC cells were smaller than the T cells, but larger than conventional exosomes and lysosomes (*de Araujo et al., 2020*; *Gurung et al., 2021*). Acquisition of CD45 by the tumor cell required contact between the TNBC cell and T cell demonstrating that the transfer of trogosomes were not a result of secreted extracellular vesicles or exosomes by the T cell (*Sheta et al., 2023*). Trogosomes also became larger over time and altered the surrounding of subcellular structures such as the nucleus and filamentous actin (Fig. 6). The identification of $CD45^-GFP-H2B^+$ and $CD45^-DiD^+$ trogosomes indicate that different membrane proteins beyond CD45 may also be transferred. Currently, it is still not known why our immunolabeling with CD45RA resulted in a staining pattern with a punctate phenotype, whereas our *in vitro* co-cultures allowed us to visualize intact spherical structures. We hypothesize that trogosomes dissociate inside the tumor cell after a certain amount of time and may either resurface on the cancer cell membrane or be degraded by the proteosome.

We also identified the presence of Hoechst dye within trogosomes, leading us to investigate the transfer of gDNA from T cells to TNBC cells. We demonstrated that DNA originating from T cells infected with either H2B-GFP or labeled with EdU were present in tumor cells after co-culture. Confocal microscopy revealed multiple trogosomes touching the nucleus of the tumor cell. One potential function of trogosomes may include shielding transferred DNA from being detected by intracellular DNA sensors, such as cGAS/STING and AIM2 (*Motwani, Pesiridis & Fitzgerald, 2019*; *Wilson et al., 2015*). DNA may then be transferred from the trogosome to the tumor cell nucleus, resulting in transcriptional changes in the tumor cell. Much like the acquisition of protein through trogocytosis, it is

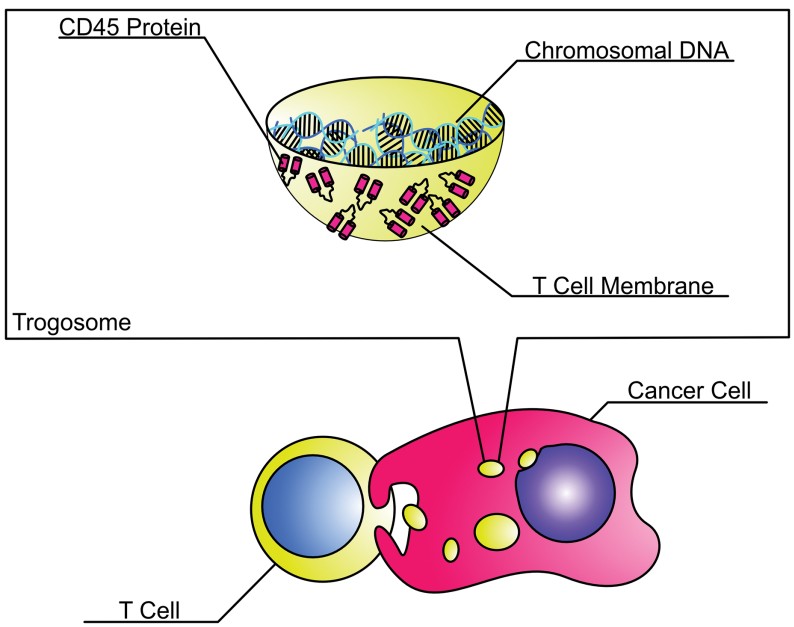

**Figure 6 Anatomy of a trogosome.** Triple negative breast cancer cells acquired trogosomes containing genomic DNA encapsulated by T cell lipid membrane and membrane protein fragments including CD45. Trogosomes are smaller than T cells but are larger than conventional exosomes and extracellular vesicles. They can also alter the shape of subcellular structures, including the nucleus and actin filaments.

still unclear whether tumor cells that acquire gDNA from T cells sustain transcription of immune cell genes such as *PTPRC*.

Trogocytic interactions have been characterized in tumor cells as early as 2015 but have largely been mostly focused on the transfer of proteins from tumor cells to immune cells, such as PD-1/PD-L1and HER2 transfer (*Hasim et al., 2022*; *Suzuki et al., 2015*). Many of these studies have also focused on trogocytosis between cancer cells that were derived from HSCs such as leukemia and healthy immune cells, two populations that canonically undergo trogocytosis with each other (*Ochs, Häusser-Kinzel & Weber, 2023*). Recently, the immunomodulatory effects of trogocytosis between cancer cell and immune cells were characterized through the transfer of TIM3 and CTLA4 from T cells to colorectal cancer cells and PD-1 transfer from leukemia cells to NK cells (*Hasim et al., 2022*; *Shin et al., 2021*). These data also suggest that the transfer of CD45 and other immune cell protein markers may also result in immunomodulation. Innate immunity relies on the recognition of foreign peptides to specify and flag cells for destruction by adaptive immunity (*Miyake & Karasuyama, 2021*). Through the expression of CD45 and other immune cell proteins, tumors may be able to avoid detection by the immune system while proliferating, differentiating, and metastasizing.

Taken altogether, these findings warrant further investigation to understand whether trogocytosis is a common feature across all subtypes of breast tumors due its prevalence in all stages of triple negative breast cancers. Although it is unclear whether there is a

difference between specific subtypes of breast tumors, trogocytosis has previously been shown to be attenuated when PI3K signaling was inhibited by using synthetic inhibitors such as Wortmannin (*Hasim et al., 2022*; *Shin et al., 2021*). Somatic mutation of *PIK3CA* gene, resulting in a constitutively active form of PI3K, is found in over 40% of primary breast cancers and may contribute to sustained trogocytic interactions between breast tumor cells and immune cells (*Martínez-Sáez et al., 2020*; *Miricescu et al., 2020*).

There has been incredible advancement for targeted therapies in hormone-positive BC and HER2-positive BC subtypes, but targeted treatment options for TNBC are lacking and the standard of care includes traditional cytotoxic chemotherapy (*Shi et al., 2024*). Although the incorporation of immunotherapy with the anti-PD-1 immune checkpoint inhibitor pembrolizumab, has improved treatment response and outcomes of TNBC, there is still a significant subset of patients who do not respond and are "immune cold"; identification of further biomarkers to increase treatment efficacy amongst non-immunogenic TNBC are urgently needed (*Schmid et al., 2020*). Cancer cell trogocytosis of immune cell protein may lower the immunogenicity of the tumor due to the acquisition of immune cell proteins and gDNA. Further investigation into the biomechanics and downstream consequences of tumor cell trogocytosis presents an opportunity to identify novel biomarkers for the treatment of TNBCs. Due to the prevalence of immune cell proteins being expressed in stage I TNBCs, identification of biomarkers may not only lead to the development of novel inhibitors and earlier detection methods of aggressive TNBCs.

## CONCLUSION

In this study, we discovered the expression of CD45 found within TNBC cells and demonstrated that the acquisition of CD45 protein can occur in a contact-dependent manner between T cells and TNBC cells through trogocytosis. We identified CD45RA, a surface membrane protein found on naïve T cells, expressed on 30–60% of all tumor cells in patient-derived FFPE samples. *In vitro* co-cultures using established breast cancer cell line models with healthy donor derived primary T cells revealed one potential origin of CD45 expression on tumor cells. We also demonstrated that CD45 was transferred alongside T cell membrane fragments in a spherical structure called a trogosome. Some trogosomes contained gDNA transferred from the T cell transferred to the tumor cell. Altogether, these data established a novel mechanism by which TNBCs acquire and express immune cell antigens extrinsic of their transcriptome. We intend to follow up these results by further characterizing trogocytosis across all three major breast cancer subtypes as well as elucidating the molecular mechanisms that drive trogocytosis in breast cancers.

## ACKNOWLEDGEMENTS

We would like to thank N. Sardinas for their review of this manuscript prior to submission and J.C. Fitch from the University of Arizona Human Immune Monitoring Facility for his assistance with flow cytometry panel design and analysis. We would also like to acknowledge J. Talaska, H. Britton, V. Smith, C. Semerad, and H. Jensen-Smith from the Advanced Microscopy Core Facility and Flow Cytometry Core Facility at the University of

Nebraska Medical Center for their technical assistance. We used the University of Nebraska Medical Center—UNMC Advanced Microscopy and Flow Cytometry Core Facilities.

### Funding

This research is supported by state funds from the Nebraska Research Initiative (NRI) and The Fred and Pamela Buffett Cancer Center's National Cancer Institute Cancer Support Grant (P30 CA036727). RRID: SCR_022467, SCR_017736, P20 GM103427 (NIGMS, NE-INBRE), P30 GM106397 (NIGMS, NCS), P20GM130447 (NIGMS, CoNDA), P30 CA036727 (NCI, Buffett Cancer Center), S10RR02730 (NIH), S10OD030486 (NIH), Nebraska Research Initiative, UNMC Vice Chancellor for Research Office. Research in this publication was also supported by the National Cancer Institute of the National Institutes of Health under award number P30 CA023074 awarded to Anutr Sivakoses. The funders had no role in study design, data collection and analysis, decision to publish, or preparation of the manuscript.

### Grant Disclosures

The following grant information was disclosed by the authors:
Nebraska Research Initiative (NRI) and The Fred and Pamela Buffett Cancer Center's National Cancer Institute Cancer Support Grant: P30 CA036727.
RRID: SCR_022467, SCR_017736.
NIGMS, NE-INBRE: P20 GM103427.
NIGMS, NCS: P30 GM106397.
NIGMS, CoNDA: P20GM130447.
NCI, Buffett Cancer Center: P30 CA036727.
NIH: S10OD030486, S10RR02730.
Nebraska Research Initiative.
UNMC Vice Chancellor for Research Office.
National Cancer Institute of the National Institutes of Health: P30 CA023074.

### Competing Interests

The findings in this study have been compiled and filed into patent: Bothwell, A., Marcarian, H. and Sivakoses, A. Trogocytosis in cancer cells and methods for treating cancer related thereto. Filed 04/05/2023. UA023-192 PRO 095364/00115. (45552618). Converted to PCT on 04/03/24.

### Author Contributions

- Anutr Sivakoses conceived and designed the experiments, performed the experiments, analyzed the data, prepared figures and/or tables, authored or reviewed drafts of the article, and approved the final draft.

- Haley Q. Marcarian conceived and designed the experiments, performed the experiments, analyzed the data, authored or reviewed drafts of the article, and approved the final draft.
- Anika M. Arias performed the experiments, authored or reviewed drafts of the article, and approved the final draft.
- Alice R. Lam performed the experiments, authored or reviewed drafts of the article, and approved the final draft.
- Olivia C. Ihedioha analyzed the data, authored or reviewed drafts of the article, and approved the final draft.
- Juan A. Santamaria-Barria conceived and designed the experiments, analyzed the data, authored or reviewed drafts of the article, and approved the final draft.
- Geoffrey C. Gurtner conceived and designed the experiments, authored or reviewed drafts of the article, and approved the final draft.
- Alfred L. M. Bothwell conceived and designed the experiments, authored or reviewed drafts of the article, and approved the final draft.

## Human Ethics

The following information was supplied relating to ethical approvals (*i.e.*, approving body and any reference numbers):

University of Arizona eIRB.

## Data Availability

The TrogoTracker v1.1 is available at GitHub and Zenodo:

- https://github.com/BothwellLab/TrogoTracker1.1.

- Sivakoses, A. (2024). TrogoTracker v1.1. Zenodo. https://doi.org/10.5281/zenodo.15008539.

The Flow Cytometric Data is available at Figshare:

- https://figshare.com/projects/Triple_negative_breast_cancer_cells_acquire_lymphocyte_proteins_and_genomic_DNA_during_trogocytosis_with_T_cells/234878.

- Sivakoses, Anutr (2025). MDAMB231 EdU Co-Cultures. figshare. Dataset. https://doi.org/10.6084/m9.figshare.28242134.v1.

- Sivakoses, Anutr (2025). MDAMB231 Co-Cultures. figshare. Dataset. https://doi.org/10.6084/m9.figshare.28242137.v1.

- Sivakoses, Anutr (2025). MDAMB436 Co-Cultures. figshare. Dataset. https://doi.org/10.6084/m9.figshare.28242155.v1.

- Sivakoses, Anutr (2025). HCC1937 Co-Culture Data. figshare. Dataset. https://doi.org/10.6084/m9.figshare.28242170.v1.

- Sivakoses, Anutr (2025). MDAMB231 GFP-H2B Co-Culture. figshare. Dataset. https://doi.org/10.6084/m9.figshare.28242173.v1.

The raw data is available in the Supplemental Files.

## Supplemental Information

Supplemental information for this article can be found online at http://dx.doi.org/10.7717/peerj.19236#supplemental-information.

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
