# Peer review of "Triple negative breast cancer cells acquire lymphocyte proteins and genomic DNA during trogocytosis with T cells"

_PeerJ, doi:10.7717/peerj.19236_

## Round 0.1 · original submission · Minor Revisions

In the manuscript entlited ¨Triple negative breast cancer cells acquire lymphocyte proteins and genomic DNA during trogocytosis with T cells¨, authors describe results supporting the hypotesis of ¨a novel mechanism by which TNBCs acquire and express immune cell antigens extrinsic of their transcriptome¨. This hypothesis is new and relevant to understanding how TNBC could escape from immunosurveillance.

Nevertheless, some issues need attention before acceptance:

1. In line 29, Abstract section: Describe the meaning of the PTPRC, CD45RA abbreviation. Verify this
2. In lines 77 to 78, clarify the histological characteristics of tissue to confirm if all samples were basal-like breast cancer basal-like (BL1 or BL2 type) or if the stage was assessed.
3. In line 98, clarify the meaning of cell marker in the ¨immune cell marker of interest¨ sentence.
4. In lines 113- 128, indicate if the CD45 detection includes primary T cell or TNBC only. Do you aspirate the primary T cell before TNBC trypsinization?
5. In line 132, the authors need to include the corresponding software version.
6. In line 140, eliminate the ¨cells¨ repetition.
7. In line 141, the authors use GADPH as a housekeeping gene for MDAMB23 because of high expression; the other two cell lines have the same high expression of GADPH.
8. In Figure 1A, include the objective used in the microscopy in the figure legend.
9. In line 209, the authors indicate the use of Transwell; nevertheless, this protocol wasn´t described in the method section. This action is mandatory: Include this information in the corresponding section.
10. Related to lines 264 to 266, including the protocol used to obtain these results in the corresponding section. Response to this suggestion is mandatory.
11. Related to lines 270 to 271, including the protocol used to obtain these results in the corresponding section. Response to this suggestion is mandatory.
12. In Supplementary Figure 1, verify if the image is rotated for the first image because it does not correspond to the other three images.
13. For the supplementary figure 3, explain why the images have no color for the first panel.
14. For all supplemental figures, include a figure legend.
15. Declares the post hoc statistical analysis realized by the data where correspond.

Include a point-by-point response for the reviewer and editor.

·

Basic reporting

The authors made use of clear, unambiguous professional language throughout the paper. Your introduction is very remarkable but needs slight modification. I suggest rephrasing the sentences in lines 36-37 and 42-43 for better understanding.
The full meaning of HER2 needs to be stated in line 52.
Your introduction needs more detail. A sentence or two on information or definition of TNBC after line 57 would be informative.
The overall aim or goal of the study needs to be stated in the Introduction section. Preferably, as the last statement in the introduction.
I applaud the authors for describing each figure. Your raw data is very descriptive and detailed, but consider citing the reference for Figure 6 in the Figure 6 description

Experimental design

The research is within the Aims and Scope of the journal. I suggest rephrasing the sentences in lines 96-98 for better understanding.

Validity of the findings

All underlying data have been provided; they are comprehensive, statistically valid, and well-controlled.

Additional comments

I suggest the sentence in lines 380-382 be checked.
I recommend you check the dates mentioned in the sentence in line 386.

Reviewer 2 ·

Basic reporting

1) The authors do not mention why they focus solely on CD45RA in the article. It may be obvious to a niche reader but not to a wider audience.

Experimental design

No comment

Validity of the findings

No comment

Additional comments

Sivakoses et al manuscript's investigates how TNBC cells undergo trogocytosis and acquire immune cell proteins and gDNA. The authors use extensive techniques in demonstrating the acquisition of the CD45 protein amongst other markers in TNBC cells. The results are presented well and the conclusions support the findings.

One comment: Why was the CCR7 protein not considered? It is also expressed by Naive T cells.

---

## Round 0.2 · accepted · Accept

This manuscript has been accepted by two independent reviewers and the editor. The authors have addressed all of the reviewers' and editors´ comments.
Please include the Conflict of Interest statement in the final manuscript once the production team contacts you.

·

Basic reporting

The authors made use of clear, unambiguous professional language throughout the paper. The modifications I highlighted in the initial review have been addressed.
The full meaning of HER2 has been stated, and a better understanding has been provided.
The authors have incorporated more details in their introduction. More information or a definition of TNBC has been provided and is informative.
The overall aim or goal of the study is stated in the Introduction section, as seen in the last statement.
I applaud the authors for describing each figure. Your raw data is very descriptive and detailed, but consider citing the reference for Figure 6 in the Figure 6 description.

Experimental design

The sentences in previous lines 96-98 (now lines 112-116) have been rephrased for better understanding.

Validity of the findings

All underlying data have been provided; they are comprehensive, statistically valid, and well-controlled.

Additional comments

Most of the concerns I highlighted were addressed by the authors.
The word 'to' is missing in the acknowledgement of Sardinas
The Conflict of Interest session is missing in the updated review. Is there a reason for that?

Reviewer 2 ·

Basic reporting

No comment

Experimental design

No comment

Validity of the findings

No comment

Additional comments

The authors have addressed my comments.